# Presence and Persistence of *Listeria monocytogenes* in the Danish Ready-to-Eat Food Production Environment

Nao Takeuchi-Storm [1,*][ID], Lisbeth Truelstrup Hansen [1][ID], Niels Ladefoged Nielsen [2] and Jens Kirk Andersen [1]

1  National Food Institute, Technical University of Denmark, 2800 Kongens Lyngby, Denmark
2  Danish Veterinary and Food Administration, Stationsparken 31, 2600 Glostrup, Denmark
*  Correspondence: naots@food.dtu.dk

**Abstract:** *Listeria monocytogenes* is an ubiquitously occurring foodborne bacterial pathogen known to contaminate foods during the production processes. To assess the presence and persistence of *L. monocytogenes* in Danish ready-to-eat (RTE) food production companies in response to a Listeria awareness campaign, the production environment of selected companies were sampled in 2016 and in 2020. Whole genome sequencing (WGS) was performed to characterize the isolates ($n = 50$, plus 35 isolates obtained from the routine surveillance during 2016–2020), including investigation of the presence of virulence, persistence and resistance genes. The number of companies that tested positive by culture was 17/39 (43.6%) in 2016 and 11/34 (32.4%) in 2020, indicating a limited effect of the campaign. WGS analyses of the 85 isolates showed that the most common sequence types (STs) were ST8 and ST121. The single nucleotide polymorphism (SNP) analysis showed that isolates coming from the same company and belonging to the same ST exhibited <10 SNP differences regardless of the sampling year and whether the samples came from the environment or products, indicating the persistence of the specific STs. Several prevalent STs were found in clinical cases concurrently, including genetically similar isolates. This highlights the issue of persistent *L. monocytogenes* in the food production environment and the need for improved risk communication and mitigation strategies.

**Keywords:** whole genome sequencing (WGS); food safety; serotype; risk management





## 1. Introduction

*Listeria monocytogenes* is an opportunistic foodborne pathogenic bacterium that can cause severe clinical disease in humans and animals (listeriosis). Despite the low number of confirmed human cases, listeriosis is considered one of the most serious foodborne diseases in the EU due to its high mortality and hospitalization rate [1]. In Denmark, there are approximately 40–60 laboratory confirmed listeriosis cases with up to five outbreaks per year [2] and its disease burden has been estimated as the third highest among seven foodborne pathogens, after campylobacteriosis and salmonellosis [3].

*Listeria monocytogenes* is a facultative anaerobic bacterium with the ability to survive and grow outside the animal host cells. It is ubiquitous in the natural environment such as in soil and water and can be transmitted to the food chain through various routes. Unlike other foodborne pathogens, *L. monocytogenes* can multiply under various "stressful" conditions such as pH ranging from 4.6 to 9.2, relatively low water activity ($a_w$ of 0.90) and low temperatures [4–6], thus enabling the bacterium to survive under various food processes and storage. Additionally, the pathogen can form biofilms on surfaces and tolerate disinfectants and antimicrobial agents [7]. Sufficient heating can eliminate the pathogen during food processing, but food products may become re-contaminated after the lethal treatment due to *L. monocytogenes* being present in the food processing environment. Ready-to-eat (RTE) food products that either receive no listericidal treatment or become re-contaminated during packing steps and have a composition that allows for growth of *L. monocytogenes* within the refrigerated shelf-life, constitute the main source of listeriosis outbreaks [7,8].

Contributing to the problem is that *L. monocytogenes* can persist in the production environment for months and years. In Denmark, use of earlier typing methods showed that the same type of *L. monocytogenes* appeared in products and the production environment in a cold-smoked salmon processing plant over a 4-year period [9]. A more recent study by Lassen et al. [10] used whole genome sequencing (WGS) to show that *L. monocytogenes* isolates taken from RTE smoked fish products and the production environment were genetically similar to clinal outbreak isolates. WGS was also used to determine that a large listeriosis outbreak (17 deaths, 41 cases) in Denmark in 2014 was caused by consumption of a contaminated RTE spiced meat roll product [11]. In response to these outbreaks, the Danish Veterinary and Food Administration (DVFA) launched a Listeria awareness campaign in 2015–2016 with information and educational activities directed towards producers of high-risk products, food inspectors and the public, including publication of a comprehensive *Listeria in Food* guide on the official DVFA website (foedevarestyrelsen.dk) directed towards consumers and producers of RTE foods. However, the extent to which the campaign had an effect, whether persistent *L. monocytogenes* isolates continue to occur in the Danish RTE food production environments and their significance in terms of clinical disease are still largely unknown.

Currently, risk management systems are in place to prevent occurrence of high levels of *L. monocytogenes* in food products in Denmark in accordance with EU Regulations (EC No 2073/2005; EC No 1441/2007) [12,13]. Producers must comply with rules that differentiate between products intended for susceptible population groups (absence in 25 g) and products intended for the general public, where products supporting growth of *L. monocytogenes* must contain less than 1 colony forming units (CFU)/25 g immediately after production and be marketed with a shelf-life duration that does not permit the pathogen reaching 100 CFU/g. For the category of products, which are stabilized against growth of *L. monocytogenes,* e.g., via adjustments to water activity, pH and/or product formulation, the rules specify that the content of the pathogen cannot exceed 100 CFU/g within the shelf-life. Actions will be taken if the microbiological criteria for food stuffs for *L. monocytogenes* (EC No 2073/2005) [12] are not met, regardless of the strain characteristics. As WGS becomes more available and more information on key virulence factors are elucidated, sub-typing information on the isolated strains could be used for microbial source tracking and can inform risk management decisions [14].

*Listeria monocytogenes* can be classified into four distinct phylogenetic lineages (I–IV) with most human infections being caused by lineages I and II. Lineage I is linked more to clinical cases, while lineage II is associated more with food products and persistence [15,16]. Multi-locus sequence typing (MLST) allows groupings and characterization of isolates based on the genetic diversity in seven housekeeping genes. In recent years, core genome MLST (cgMLST) schemes, i.e., genetic comparison of a shared subset of genes occurring in all strains, have also been developed to perform the strain typing with higher discriminatory power. Recent investigations have, however, shown that cgMLST is less able to differentiate between the isolates of different origins, compared to whole genome MLST or single nucleotide polymorphism (SNP) analyses utilizing all the genetic information in strains [17].

Isolates with presence of premature stop codons (PMSC) in the internalin A gene (*inlA*) have been shown to be less pathogenic, as *inlA* plays a crucial role in invasion of intestinal epithelial cells [18,19]. Harbourage of the pathogenicity islands (LIPI)-1, LIPI-3 and LIPI-4, which contain other major virulence genes, has been associated with hypervirulence. It is possible that strains could be ranked according to lineage and presence of full-length *inlA* and pathogenicity islands and be evaluated differently under risk assessment and management [20,21].

The aim of the study was to obtain a deeper knowledge of the presence and persistence of *L. monocytogenes* in the Danish RTE food production environment during 2016 and 2020 following the Listeria awareness campaign by the DVFA. Swab samples from the production environment in food companies producing the known risk products for *L. monocytogenes* were collected and analysed for the presence of *L. monocytogenes* in 2016 and 2020. Resulting isolates, as well as isolates obtained from the same companies during the DVFA's routine 2016–2020 surveillance program, were characterized by WGS to assess the strain

diversity, persistence of genetically similar strains (here defined as strains with <20 SNP differences) over the years and presence of genes encoding for antimicrobial/biocide resistance and virulence factors. Finally, putative links between the isolates from the production environment and human cases were surveyed to elucidate the significance of presence and persistence of *L. monocytogenes* in the production environment in response to the Listeria awareness campaign.

## 2. Materials and Methods

### 2.1. Bacterial Isolates and Other Relevant Information

Food production companies, which produce known *L. monocytogenes* risk products (RTE fish and meat products) in different regions of Denmark, were selected for the study in 2016 based on information from the DVFA surveillance program. The sampling design aimed for recruitment of 20 fish and 20 meat production companies. The project was repeated in 2020 with the aim of taking samples from the same companies. A replacement company was selected for sampling in 2020 if the company from 2016 was not available.

Ten swab samples from the food production environment (equipment and surfaces) were taken from each of the food companies. Sampling was carried out by experienced samplers (official food inspectors), according to the accredited procedure *Guidelines on sampling the food processing area and equipment for the detection of Listeria monocytogenes Version 3—20 August 2012* [22]. Product contact and non-contact surfaces, including from the drains, were represented from all the companies. Swab samples were sent to the DVFA's accredited laboratory (Ringsted, Denmark) and processed according to EN/ISO 11290-1 method for identification of *L. monocytogenes* by culture. One isolate from each positive sample was selected for WGS and stored frozen (−80 °C) in Tryptone Soy Broth (Oxoid, Fisher Scientific, Kamstrup, Denmark) supplemented with 30% glycerol.

In addition to this study's environmental sampling, the joint DVFA and DTU archive was searched for *L. monocytogenes* isolates, which came from products and environmental swabs collected from the same companies as part of routine surveillance in the period 2016–2020. Only the isolates with available WGS data were included in the analyses.

### 2.2. Whole Genome Sequencing

DNA extraction followed the standard DVFA protocol. Briefly, *L. monocytogenes* were revived from frozen stock (−80 °C) by streaking on Tryptic Soy blood agar (Statens Serum Institut, Copenhagen, Denmark) for 24 h at 37 °C. Single colonies were sub-cultured in Tryptic Soy Broth (Oxoid, Fisher Scientific, Roskilde, Denmark) for 18 h at 37 °C. DNA was extracted from this culture using the Easy-DNA kit (Invitrogen, Waltham, MA, USA). DNA concentrations (0.18–0.28 ng/µL) were measured using the Qubit dsDNA BR Assay Kit (Invitrogen). Sequencing was done using MiSeq sequencing machines (Illumina Inc., San Diego, CA, USA) and the Nextera XT Library preparation protocol for paired-end reads of 250 bp. The WGS data were then processed using an internal quality-control (QC) pipeline at the Center for Genomic Epidemiology, Technical University of Denmark. This pipeline performs trimming of low-quality and adaptor sequences and de novo assembly using SPAdes.

The assembled sequences were analysed by in silico bioinformatics tools available from the Center for Genomic Epidemiology (http://www.genomicepidemiology.org (accessed on 16 April 2021)). Species were identified by KmerFinder v.3.2 [23–25] and the MLST typing was performed by MLSTFinder v.2.0 [26], which uses the PubMLST database (https://pubmlst.org/ (accessed on 19 April 2021)). The assembled sequences were also uploaded to the Listeria PasteurMLST database (https://bigsdb.pasteur.fr/ (accessed on 30 July 2021)) for lineage and serogroup data.

### 2.3. Characterization of Isolates in Terms of Virulence, Resistance and Plasmids

The assembled sequences were screened for the presence of genes encoding antimicrobial and other resistance genes (stress, heat and biocide) using AMRFinderPlus v.3.10.14 with a default setting of minimum coverage of 90% and minimum identity of 90% [27]. In

addition, eight genes encoding for biocide resistance (*Tn*6188 (which includes *qacH* and *tetR* genes), *qacA*, *qacC*, *qacH*, *bcrABC*, *emrC*, *emrE* and *mdrL*) were manually searched using ABRicate with minimum 70% coverage and 90% identity [28].

Virulence elements (*inlA*, LIPI-1, LIPI-3 and LIPI-4) were also searched using ABRicate with minimum 70% coverage and 90% identity. Identification of PMSC and internal deletions were subsequently assessed by Geneious Prime v.2021.2.2 (Biomatters, Inc., Auckland, New Zealand). Using ABRicate, PlasmidFinder v.2.1 [29,30] and ResFinder v.4.1 [30,31] were run to identify and characterise the presence of known plasmid types. Coverage and identity below 90% were not considered hits and were discarded. The list of reference genes and genomes is summarized in Supplementary Table S2.

*2.4. SNP Analyses*

Single nucleotide polymorphisms (SNP) calling and construction of SNP matrices were performed using the CSIPhylogeny v.1.4 [32], where the paired-end trimmed reads were mapped to the *L. monocytogenes* EGD-e chromosome, complete genome (NC_003210.1) as a reference. The SNPs were selected with the following criteria: (1) a minimum of 10% relative depth at SNP positions, (2) a minimum distance of 10 bps within each SNP, (3) SNP quality of >30 and (4) mapping quality of >25. Phylogenetic trees based on SNP matrices were visualized by iTOL v.6 [33].

*2.5. Human Listeriosis Cases during the Study Period*

*Listeria monocytogenes* is a notifiable human disease in Denmark. Regional, national and international food and water borne outbreaks are monitored and investigated by the Central Outbreak Group (DCUG) in Demark. The group is made up of members from the Statens Serum Institute (SSI), the Danish Veterinary Food Administration (DVFA) and National Food Institute (DTU) and meets once a week to ensure a coordinated effort to manage outbreaks. *Listeria monocytogenes* isolates from patients are handled by SSI for typing, while sampling and sequencing of food and production environment are conducted jointly by DVFA and DTU, as described in Lassen et al. [10]. Outbreaks are reported to EFSA and in the Annual Report on Zoonoses. Possible links between the isolates from this study and human cases were searched by screening these public sources [2,34–37].

**3. Results**

*3.1. Summary of the Bacterial Isolates*

A total of 777 environmental samples were obtained from 53 companies, of which 32 and 20 samples were positive by culture for *L. monocytogenes* in 2016 and 2020, respectively. An overview of the samples and companies are shown in Table 1, with more details provided in Supplementary Table S1. Prevalence of *L. monocytogenes* in environmental samples identified by culture were 32/426 (7.5%) in 2016 and 20/351 (5.7%) in 2020 and the number of companies, which had at least one positive sample by culture were 17/39 (43.6%) in 2016 and 11/34 (32.4%) in 2020. A total of 20 companies (9 meat and 11 fish) were sampled both in 2016 and 2020 and 9/20 (45.0%) and 7/20 (35.0%) companies had at least one positive isolate in 2016 and 2020, respectively. Among these, four companies tested positive in both years (Supplementary Table S1).

**Table 1.** Number of samples and participating companies for sampling of food processing environment for *Listeria monocytogenes* in 2016 and 2020.

| Year | 2016 | | 2020 | |
|---|---|---|---|---|
| **Product Type of the Company** | **Meat** | **Fish** | **Meat** | **Fish** |
| No. of samples | 251 | 175 | 181 | 170 |
| No. of companies | 22 | 17 | 18 | 16 |
| No. of samples per company | | 6–25 | | 10–20 |
| No. of samples positive for *L. monocytogenes* by culture | 15 (6.0) [a] | 17 (9.7) | 10 (5.5) | 10 (5.9) |
| No. of companies positive for *L. monocytogenes* by culture | 8 (36.3) | 9 (52.9) | 7 (38.9) | 4 (25.0) |

[a] Numbers in brackets are % positives.

Of the 52 *L. monocytogenes* isolates, the quality of WGS data from two isolates were found to be suboptimal. The remaining 50 *L. monocytogenes* isolates (30 from 2016 and 20 from 2020), which came from 24 companies, were used for further WGS analyses (Table 2 and Table S1).

**Table 2.** Identification of additional *Listeria monocytogenes* isolates from the production environment and products from the 53 selected food production companies. Isolates were obtained during routine surveillance performed by the Danish Food and Veterinary Administration during 2016–2020.

| | 2017 | | 2018 | | 2019 | | 2020 |
|---|---|---|---|---|---|---|---|
| Sample source | Env | Prod | Env | Prod | Env | Prod | Prod |
| No. isolates with WGS data | 6 | 4 | 2 | 3 | 6 | 5 | 9 |
| No. companies with *L. monocytogenes* | 2 | 3 | 1 | 1 | 2 | 4 | 3 |

Abbreviations: Env = environment, Prod = product. No additional strains were identified for 2016 and from the production environment data in 2020.

Additional database search for *L. monocytogenes*, which had been isolated during the DVFA's routine surveillance of the participating companies in 2016–2020, resulted in 35 additional isolates (21 product and 14 environmental, Table 2), none of which came from the 2016 surveillance program. In total, 85 isolates with WGS data from 27 companies were used for further WGS analyses.

### 3.2. Distribution of STs and SNP Analyses

The metadata of the 85 isolates and selected output from the Listeria PasteurMLST database are shown in Supplementary Table S3. Sixteen STs of lineages I and II were identified; ST1, ST6, ST87, ST296 and ST416 belong to lineage I and the rest to lineage II (Table 3). An overview of the isolates and their origin, i.e., company, sampling year and source (environment vs. product) and STs are summarised in Figure 1. The STs, which were isolated over two or more years from the same company were: ST7, ST8, ST121, ST399 and ST451. Ten out of 27 companies harboured isolates of different STs (Table 3).

**Table 3.** Overview of the STs of 85 *Listeria monocytogenes* isolates obtained from the food processing environment and products between 2016 and 2020.

| ST | Total No. Isolates | Lineage | Company ID | Company Type | No. Isolates | ST | Total No. Isolates | Lineage | Company ID | Company Type | No. Isolates |
|---|---|---|---|---|---|---|---|---|---|---|---|
| ST1 | 2 | I | 9<br>18 | Meat<br>Fish | 1<br>1 | ST37 | 4 | II | 4<br>20<br>22 | Fish<br>Meat<br>Meat | 1<br>1<br>2 |
| ST6 | 2 | I | 3<br>22 | Meat<br>Meat | 1<br>1 | ST87 | 1 | I | 20 | Meat | 1 |
| ST7 | 3 | II | 27 | Meat | 3 | ST121 | 19 | II | 2<br>11<br>16<br>17<br>21<br>23<br>24 | Fish<br>Fish<br>Meat<br>Meat<br>Meat<br>Meat<br>Fish | 11<br>1<br>2<br>1<br>1<br>1<br>2 |
| ST8 | 24 | II | 1<br>4<br>5<br>8<br>12<br>20<br>26<br>27 | Fish<br>Fish<br>Meat<br>Meat<br>Meat<br>Meat<br>Meat<br>Meat | 1<br>11<br>2<br>3<br>1<br>2<br>2<br>2 | ST155 | 3 | II | 16<br>26 | Meat<br>Meat | 2<br>1 |

**Table 3.** *Cont.*

| ST | Total No. Isolates | Lineage | Company ID | Company Type | No. Isolates | ST | Total No. Isolates | Lineage | Company ID | Company Type | No. Isolates |
|---|---|---|---|---|---|---|---|---|---|---|---|
| ST9 | 4 | II | 4<br>7<br>10 | Fish<br>Meat<br>Meat | 1<br>2<br>1 | ST296 | 1 | I | 14 | Meat | 1 |
| ST14 | 4 | II | 17<br>26 | Meat<br>Meat | 1<br>3 | ST399 | 4 | II | 13<br>18<br>24 | Meat<br>Fish<br>Fish | 1<br>2<br>1 |
| ST18 | 2 | II | 9<br>25 | Fish<br>Fish | 1<br>1 | ST416 | 1 | I | 5 | Meat | 1 |
| ST21 | 2 | II | 6<br>27 | Meat<br>Meat | 1<br>1 | ST451 | 9 | II | 15 | Meat | 9 |

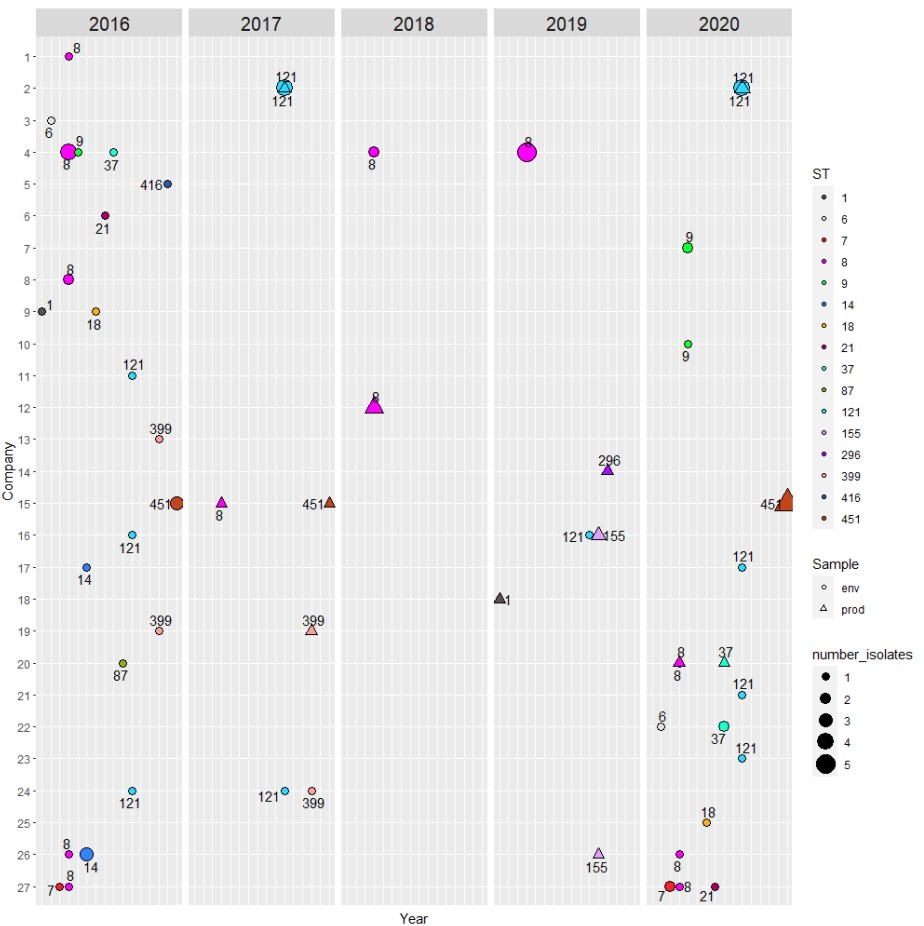

**Figure 1.** Overview of the number of *L. monocytogenes* isolates from the participating companies according to different STs and sampling year. Only the companies (27/53) that harboured at least one positive isolate either in product (prod) or environmental (env) samples are shown.

The phylogenetic tree based on SNP analyses showed clear separation according to lineages and STs (Figure 2). The isolates belonging to the same ST and coming from the same company showed only small SNP differences (<10 SNP) regardless of sampling year and source (environment vs. product), e.g., ≤6 SNP differences among the nine ST451 isolates. Isolates which belonged to the same ST but came from different companies showed greater SNP differences, generally >50. In the case of ST121 (Figure 3a), a closer look revealed that company no. 2 harboured *L. monocytogenes* isolates belonging to this ST,

which exhibited between 0 and 9 SNP differences regardless of their year of isolation (2017 and 2020). ST8 (Figure 3b) did not show as clear clustering among the isolates from within a company, i.e., isolates from company no. 4 exhibited between 1 to 55 SNP differences, while the SNP difference between company no. 4 and no. 12 was only 34.

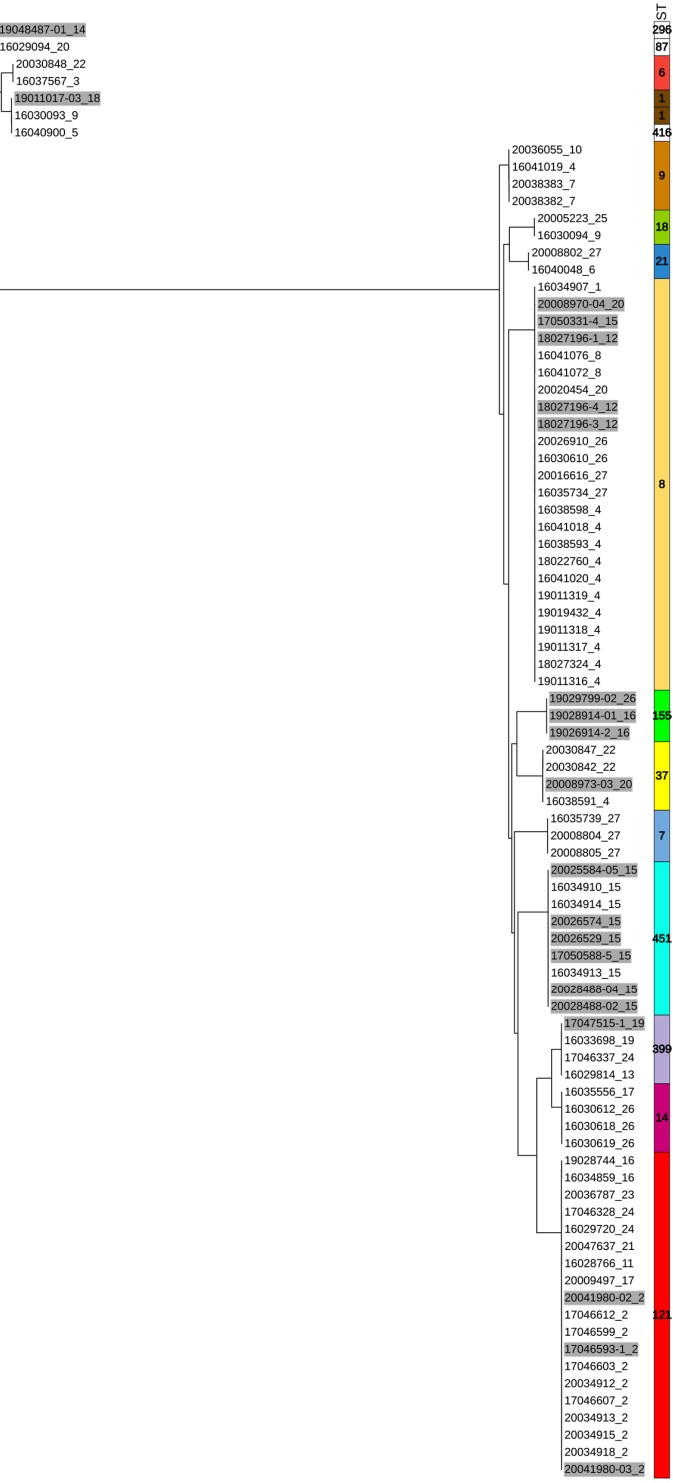

**Figure 2.** Phylogenetic tree of 85 *Listeria monocytogenes* isolates taken from food production environment and products of 27 companies between 2016 and 2020, based on SNP analyses. The first two digits indicate the sampling year (16–20) and the last two digits indicate the company ID; no highlight and grey highlight indicate environmental and product samples, respectively; STs are showed in the right column with colour.

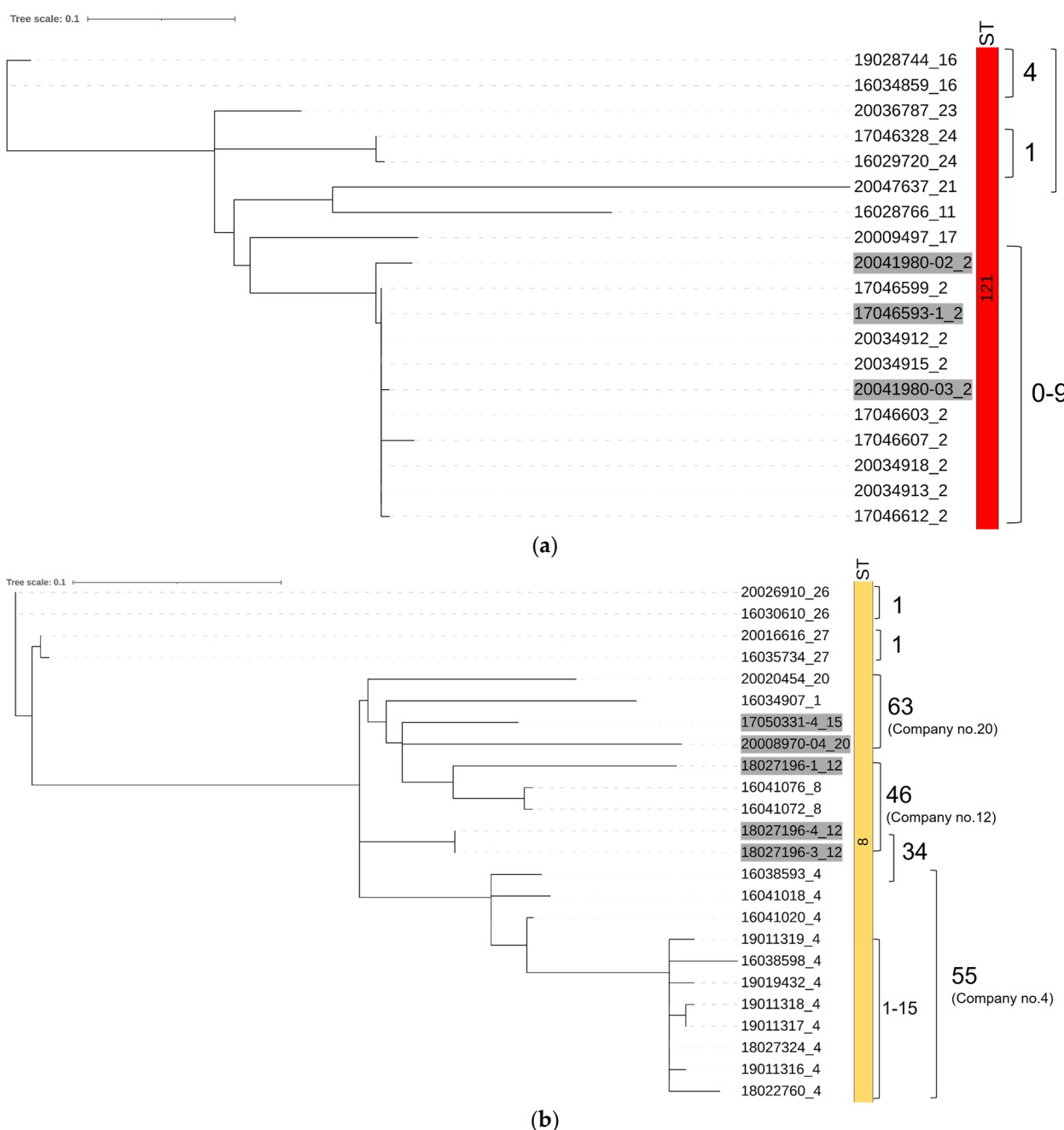

**Figure 3.** SNP-based phylogenetic trees and SNP-differences of *L. monocytogenes* isolates belonging to (**a**) ST121 and (**b**) ST8. The first two digits indicate the sampling year (16–20) and the last two digits indicate the company ID; no highlight and grey highlight indicate environmental and product samples, respectively; STs are showed in the right column with colour; and SNP differences between selected isolates are shown as numbers on the right.

### 3.3. Characterisation of the Selected Virulence, Resistant Genes and Plasmids

The presence of selected virulence and resistance genes as well as plasmids corresponded well with the STs (Figure 4). Internalin A was found in all isolates except for one isolate, which is likely to have contained the gene but only showed 62.6% coverage due to the location at the start of a contig (1503 bp). Three of the ST9 isolates had PMSC mutation type 29 (allele *inlA*_47), while the last isolate contained PMSC type 11 (allele *inlA*_44) according to the current typing overview of reported PMSCs in *inlA* [38–40]. All ST121 isolates contained PMSC mutation type 6 (allele *inlA*_49) [41]. ST6 isolates were

found to have a deletion of 9 bp from positions 2212–2220 in the gene, which may or may not produce a functional *inlA*.

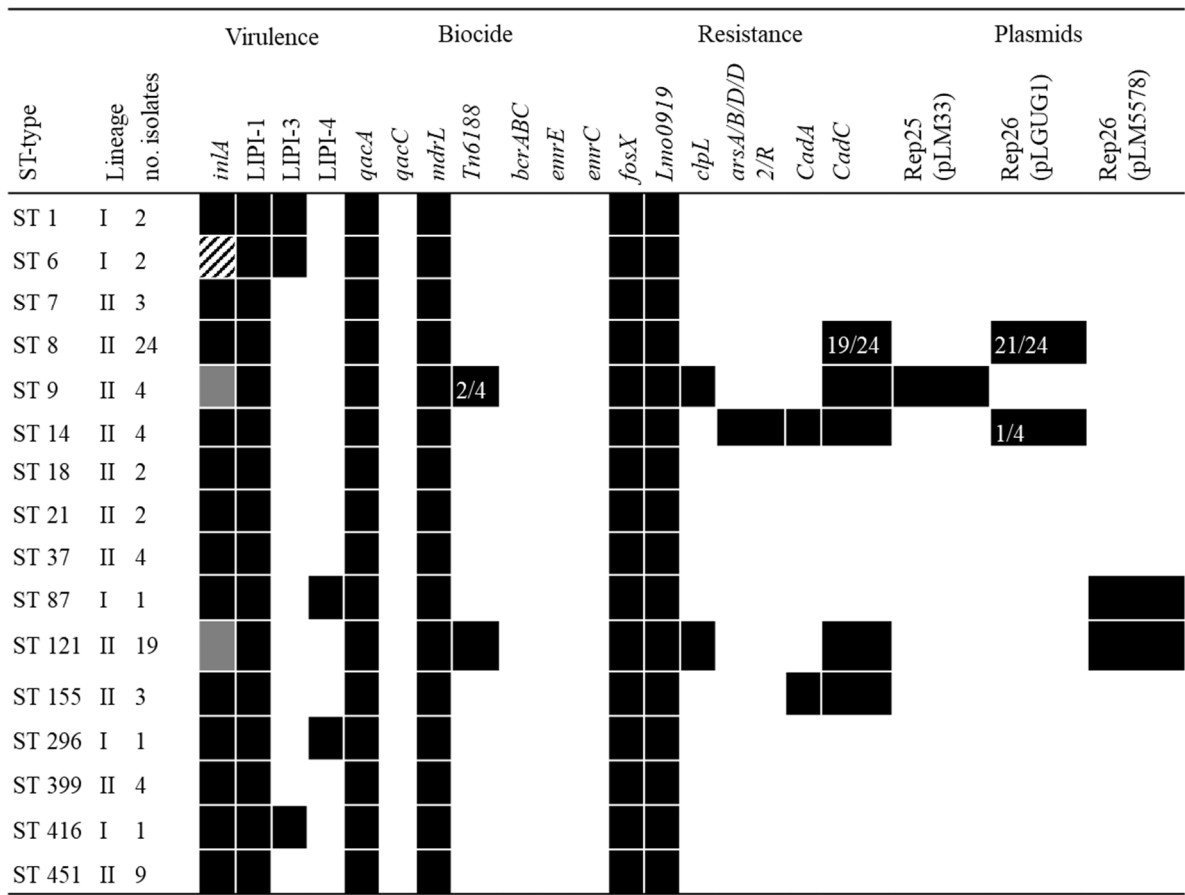

**Figure 4.** Presence of selected elements and genes associated with virulence, biocide and antibiotic resistance and plasmids in the isolates belong to the different STs. Gray cells in internalin A (*inlA*) indicate presence of premature stop codons and diagonal shading indicates inconclusive result. The numbers in cells indicate the number of isolates which contained the genes out of the total number of isolates.

LIPI-1 was present in all isolates, while LIPI-3 was found in five isolates of ST1, ST6 and ST416, all of which belonged to lineage I. LIPI-4 was found in the remaining lineage I strains (ST87 and ST296).

Out of the eight genes for biocide resistance investigated, *qacA*, *mdrL* were found in all isolates. *Tn6188*, which includes *qacH*, was found in 21 isolates (all isolates from ST121 and 2 of ST9). No isolates were found to contain *qacH* resistance genes without the concurrent presence of *Tn6188*. None of the other genes, which confer tolerance to quaternary ammonium compounds (QACs) (*qacC*, *bcrABC*, *emrC*, *emrE*), were found.

All isolates contained resistant genes against fosfomycin (*fosX*) and lincosamide (*lmo0919*). ATP-dependent protease (ClpL)-encoding *clpL* gene, which increases heat resistance in *L. monocytogenes*, was found in ST 121 and ST 9 isolates. The *clpL* gene was on the same contig as the replicon genes of pLM5578 (except for one of ST121 isolate) and pLM33 of ST9 isolates, suggesting that the *clpL* gene is likely to be encoded on the pLM5578 and pLM33 plasmids. Likewise, many of the *cadC* genes were found to be on contigs that were associated with known plasmid genes. However, *cadC* genes were also identified in isolates belonging to ST14 and ST155, while no plasmid replicons were found in these isolates.

*3.4. Possible Links to Danish Human Listeriosis Cases*

According to published reports and communication with DCUG, 10 different STs had been implicated in human cases in Denmark during the study period (Table 4). Five of these STs were also found in the studied RTE food companies.

**Table 4.** *Listeria monocytogenes* outbreak isolates (ST and source) as reported in the Food- and Waterborne Outbreak Database published in the Annual Report on Zoonoses from 2016–2020 [2,34–37].

| Year | ST Type (Lineage [a]) | Source |
|------|------------------------|--------|
| 2016 | ST4 (I) | Cold cuts of meat |
| 2016 | ST6 (I) | Unknown |
| 2017 | ST1 (I) | Unknown |
| 2017 | ST1247 (II) | Unknown |
| 2017 | ST8 (II) | Imported smoked salmon |
| 2017 | ST55 (I) | Unknown |
| 2017 | ST6 (I) | Imported sweet corn |
| 2018 | ST20 (II) | Unknown |
| 2018 | ST8 (II) | Unknown |
| 2019 | ST1 (I) | Salads |
| 2020 | ST7#7 (II) | Unknown |
| 2020 | ST394#1 (II) | Hot-smoked trout |
| 2020 | ST451#2 (II) [b] | Hot-smoked fish products |

[a] Lineage information has been added by searching the ST type in PasteurMLST database (https://bigsdb.pasteur.fr/cgi-bin/bigsdb/bigsdb.pl?db=pubmlst_listeria_seqdef (accessed on 15 May 2022)). [b] This outbreak strain caused two additional cases in 2014.

A closer look at the WGS from human ST451 isolates from 2014 and 2020 showed that there were <10 SNP differences between isolates from these clinical cases and food production isolates from the present study. Moreover, isolates taken from the environment in 2016 were genetically nearly identical (≤6 SNPs) to those collected from the products in 2020. Taking that information together, it seems plausible that this specific *L. monocytogenes* ST451 strain has persisted over time in the production environment and may have contributed to human cases, with a 6-year interval.

## 4. Discussion

In the present study, *L. monocytogenes* were found from the production environment of 17/39 (43.5%) companies in 2016 and 11/34 (32.4%) in 2020, indicating that the prevalence was slightly lower in 2020 but that *L. monocytogenes* was still found in one third of the participating RTE food processing facilities. It should be noted that the difference in prevalence between the two years cannot be compared directly since only 20 companies participated in both years and the number of samples per company were variable. Additionally, sampling in 2020 was more focused in the area after heat treatment rather than the area where the raw materials were handled. This may have resulted in the lower number of positives in 2020.

*L. monocytogenes* isolates belonging to 16 different STs were found in the environmental and food product samples from the RTE food companies (Figure 1). In Norway, STs belonging to clonal complex (CC) 7, CC8, CC9, CC121 were the most prevalent [16] and, likewise, CC9 and CC121 were the most frequently found strain types in Switzerland [42]. In our study, ST8 and ST121 were the most prevalent isolates and ST7 were found in lesser extent, but no isolates belonging to CC9 (e.g., ST9 and ST477) were found. It has been hypothesized that ST9 strains are highly adapted to meat production environments [17,42]. Despite our limited sample size, ST9 was in low prevalence in the Danish meat production environment. While each company was sampled several times, it is possible that all STs were not captured since WGS were performed on only one of the colonies taken randomly from the culture plates from each positive sample. It should be noted that selective media may also select for specific STs during culture [43].

Results from the present study showed that the same STs can be isolated over long periods from the same companies; the isolates belonging to the same ST and obtained from the same companies were genetically similar (<10 SNP) regardless of the sampling years or whether the samples were taken from the products or the environment (Figure 3). Wang et al. [44] have shown that the probability of two isolates with <20 SNP differences coming from the same facility is high (70%) and suggested 20 SNP to be used as a threshold for calling isolates identical in *L. monocytogenes* outbreak and traceback investigations. On the contrary, Fagerlund et al. [17] found small SNP differences (7–11 SNPs) between two isolates from two different facilities, making it difficult to deduce the origin of an outbreak unless the two facilities were linked through an unrealized common source of contamination. Supplementation with wgMLST analyses would provide further information, as wgMLST includes accessory genes, while SNP analyses exclude variable elements such as plasmids and prophages [17,45]. However, this was beyond the scope of the present study. The isolates belonging to the same ST and coming from the same companies in our study clustered together, while larger SNP differences were found among the ST isolates taken from different companies (Figures 2 and 3a). ST8 was an exception and will be discussed further below. Taken together, the low number of SNP differences found among isolates from individual RTE food companies in the present study is a strong indicator that *L. monocytogenes* isolates have persisted in the production environment over time, and transfer between the environment and food products occurred during the production.

While isolates of the same STs from different companies showed generally larger SNP differences (>50 SNP), highly similar (34 SNPs) ST8 isolates were obtained from companies no. 4 and 12 (Figure 3b). Interestingly, ST8 isolates from company no. 4 showed large genetic variation (up to 55 SNP), making it difficult to evaluate if the ST8 isolates persisted and mutated over time, or there were repeated introductions of ST8 *L. monocytogenes* into the RTE food company. A recent Norwegian study also reported finding genetically similar isolates belonging to STs such as ST451 and ST37 in different natural environments, geographical locations and clinical cases [46], indicating that our knowledge on the transmission of the bacterium through the natural and urban environment is limited.

Currently, all STs of *L. monocytogenes* are considered as pathogenic and control measures are taken as soon as *L. monocytogenes* are found. However, there seem to be variations among different strains in terms of pathogenicity and persistence and more genes that are suspected to play an important role for virulence have been identified in recent years. A recent report from FAO/WHO [20] has proposed sub-type specific risk assessments for virulence ranking of *L. monocytogenes* strains. The ranking is based on the information on lineage and presence/absence of LIPI-1/3/4 and *inlA* PMSC, with proposed grouping as follows (highest to lowest risk): (1) Lineage I strains with LIPI-1, 3, or 4 with a full length *inlA*; (2) strain of any lineage with complete and functional LIPI-1 and with full length *inlA*; and (3) any *L. monocytogenes* strain with a truncated *inlA*. The proposed ranking is still in its infancy and not validated. However, if it is optimised, validated and implemented, this ranking could potentially lead to optimisation of resource allocations in risk management and less food waste. Considering our dataset as an example, the truncated *inlA* were present in only 23 out of 85 isolates (ST9 and ST121), making categorizing them as the lowest risk ranking group 3, while the rest are classified as either risk ranking 1 (4 STs) or 2 (9 STs). Highly similar ST451 (lineage II) isolates, which belong to the suggested risk ranking group 2, were putatively associated with concurrent human cases in our example. Although lineage I is considered as more virulent and risk ranking is higher compared to other lineages, lineage II strains do still cause sporadic clinical cases, as shown in Table 4. More studies on virulence genes, validation of the virulence gene expressions, as well as the genes' associations within the sub-types are needed to evaluate the usefulness of the proposed ranking system.

All the isolates, which were present in the RTE food companies over two or more years, belonged to lineage II in the present study. In addition, tolerance to quaternary ammonium compounds (QACs) conferred by *qacH* gene were found in the isolates with

apparently reduced virulence or truncated *inlA*, which are considered lower risk according to the proposed risk ranking. This seems to support the common notion that persistent strains are generally equal to or less virulent than non-persistent strains [21,47]. Some of the isolates, which were present over two or more years e.g., ST451, did not seemingly have any additional biocide and resistant genes than *qacA* and *mdrL* genes. However, admittedly our gene search was not extensive and other genetic mechanisms may be at play. There is still limited knowledge related to the mechanisms of persistence and virulence among different strains and such knowledge will be useful in terms of risk assessment and management of *L. monocytogenes* in the food production environment.

Results from the present study are in agreement with previous studies [17,48] that *L. monocytogenes* STs with few SNP differences are able to colonise RTE food companies, thereby demonstrating the need for a multi-facetted approach to manage the risk of *L. monocytogenes* in RTE products. The DVFA's Listeria awareness campaign launched in 2015 was a risk communication strategy aimed at reducing the prevalence and outbreaks caused by the bacterium through education of food producers, food inspectors and consumers, including susceptible segments of the population. Despite this campaign, our study shows that the overall prevalence of *L. monocytogenes* in high risk RTE facilities was largely unchanged between 2016 and 2020. Moreover, cases of listeriosis in Denmark are not going down, with recorded cases having increased from 39 in 2016 to 86 in 2022 (median 58, equivalent of 1 case per 100.000 inhabitants) [49]. Difficulties in reaching specific target groups and finding the best way to communicate the risk of listeriosis to different consumers is well documented [50]. In this review, it was also noted that while the internet is the most common way to communicate food safety information, it may not be the best medium, depending on the target group. The DVFA's campaign relied in part on the internet. It appears that new ways to reach the industry and consumers are needed to communicate about topics such as the importance of consumption prior to the best-before dates for RTE products; industrial and domestic hygiene; suitable HACCP plans; and application of *L. monocytogenes* growth controlling hurdles in the form of listericidal treatments (e.g., heat treatment), or (re)formulation of products (e.g., addition of organic acids; see Dalgaard & Mejlholm [51]).

## 5. Conclusions

The study presented some evidence for persistence of *L. monocytogenes* in the RTE food production environment. Transfer of *L. monocytogenes* between the environment and products seems to occur during food production, as isolates with the same ST from the same company were genetically almost identical (<10 SNP) regardless of sampling year and whether the isolates came from products or production environment. Clinical isolates genetically similar to the persistent strains isolated from the food production environment were identified. Prevalence of *L. monocytogenes* remained comparable between 2016 and 2020 samplings, which, taken together with the increasing trend in listeriosis cases in Denmark, may indicate that the current risk communication strategy is not working, despite the DVFA's intensive Listeria awareness campaign and availability of comprehensive information resources on the agency's website. Risk assessment and management of *L. monocytogenes* require a multi-actor approach. To optimise eradication, control and risk identification, further research is needed in terms of risk communication strategies to reach stakeholders, as well as mitigation strategies, which are based on an understanding of the mechanism of persistence and virulence.

**Supplementary Materials:** The following supporting information can be downloaded at: https://www.mdpi.com/article/10.3390/hygiene3010004/s1, Table S1: Summary of participating companies and the number of samples for the study; Table S2: List of databases and reference genes used for the study; Table S3: Metatdata of the isolates used for the study with the selected output from the Listeria PasteurMLST database.

**Author Contributions:** Conceptualization, N.L.N. and J.K.A.; methodology, N.T.-S., L.T.H., N.L.N. and J.K.A.; formal analysis, N.T.-S.; investigation, N.L.N. and J.K.A.; visualization, N.T.-S. and L.T.H.; writing—original draft preparation, N.T.-S.; writing—review and editing, N.T.-S., L.T.H., N.L.N. and J.K.A.; funding acquisition, N.L.N. and J.K.A. All authors have read and agreed to the published version of the manuscript.

**Funding:** The work was supported by the Danish Veterinary and Food Administration [project numbers: 2015-28-61-00357 and 2019-28-61-00176].

**Institutional Review Board Statement:** Not applicable.

**Informed Consent Statement:** Not applicable.

**Data Availability Statement:** Data is contained within the article or Supplementary Material. The assembled genomes used in this study are openly available in the Listeria PasteurMLST database (https://bigsdb.pasteur.fr/) with ID: BIGSdb_20210730163540_479997_44747.

**Acknowledgments:** Mirena Ivanova from National Food Institute is thanked for her assistance in WGS analyses.

**Conflicts of Interest:** The authors declare no conflict of interest. The funder was part of designing the study; collection, analyses and interpretation of data; writing of the manuscript; and decision to publish the results.

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
