# Peer review of "Presence and Persistence of Listeria monocytogenes in the Danish Ready-to-Eat Food Production Environment"

_2673-947X, doi:10.3390/hygiene3010004_

Round 1

Reviewer 1 Report

a wonderful work!

please see line 127 - verify "Scientic" - maybe should be Scientific

Reviewer 2 Report

 Comments to authors

Journal: Hygiene

Title:Presence and persistence of Listeria monocytogenes in the Danish ready-to-eat (RTE) food production environment. The research is interesting, and the manuscript looks good too but still, I have fewer minor questions as mentioned below.

Minor comments:

1.     The article needs to be revised thoroughly by English experts as it contains many syntax and grammatical errors.

2.     No. of positive samples and companies decreased in 2020 as compared to 2016. Explain the possible reasons.

3.     Line 214: Table number mistake (S3).

4.     Line 272: word ‘hits’. Change it with another suitable word.

5.     Line 281: word “Searches” is not suitable according to the sentence.

6.     Line 285: “only very small” is not suitable according to the sentence.

Major comments

1.     Highlight the key findings of your research in 4-5 points.

2.     The total number of companies and the positive by culture are not clear according to the total number is varying every time. How will you elaborate these results when the total number of companies are changing?

Reviewer 3 Report

I read with interest the paper “Presence and persistence of Listeria monocytogenes in the Danish ready-to-eat (RTE) food production environment

Authors report the results of a double sampling plan in 2016 and 2020 from the production environment in food companies: swab samples were analysed for the presence of L. monocytogenes and isolated strains were characterized by WGS and compared (diversity, genetical similarity, presence of antimicrobial encoding genes and virulence factors).

The manuscript sounds scientifically very interesting and written in a professional way. The research is appropriately described, and laboratory methods are adequately described or cited. Results are clearly presented.

I congratulate with authors for their interesting, well presented, and complete work.

Reviewer 4 Report

The manuscript entitled 'Presence and persistence of Listeria monocytogenes in the Danish ready-to-eat (RTE) food production environment' provides interesting findings. Overall, the research is well framed and executed, manuscript is clearly presented and merits publication. However, the sampling methodology in terms of the persistence of infection may be explained in detail.
